# Vaccination Coverage against Tetanus, Diphtheria, Pertussis and Poliomyelitis and Validity of Self-Reported Vaccination Status in Patients with Multiple Sclerosis

**DOI:** 10.3390/jpm12050677

**Published:** 2022-04-23

**Authors:** Silvan Elias Langhorst, Niklas Frahm, Michael Hecker, Pegah Mashhadiakbar, Barbara Streckenbach, Julia Baldt, Felicita Heidler, Uwe Klaus Zettl

**Affiliations:** 1Department of Neurology, Neuroimmunology Section, Rostock University Medical Center, Gehlsheimer Str. 20, 18147 Rostock, Germany; niklas-frahm@gmx.de (N.F.); michael.hecker@rocketmail.com (M.H.); pegah.mashhadiakbar@uni-rostock.de (P.M.); babswehr@web.de (B.S.); julia.baldt@outlook.de (J.B.); uwe.zettl@med.uni-rostock.de (U.K.Z.); 2Department of Neurology, Ecumenic Hainich Hospital, Pfafferode 102, 99974 Mühlhausen, Germany; f.heidler@oehk.de

**Keywords:** multiple sclerosis, vaccination coverage, vaccination behavior, vaccination status self-assessment, tetanus, diphtheria, pertussis, poliomyelitis

## Abstract

Multiple sclerosis (MS) is a chronic immune-mediated disease with a neurodegenerative component of the central nervous system. Immunomodulatory therapy can increase the risk of infection, which is a particular risk for MS patients. Therefore, a complete vaccination status is of utmost importance as protection against vaccine-preventable infectious diseases. Our aim was to investigate the vaccination status, vaccination card knowledge and the vaccination behavior of MS patients with regard to vaccinations against tetanus, diphtheria, pertussis and poliomyelitis. Three hundred twenty-seven patients with MS were evaluated by anamnesis, clinical examination, structured interview and vaccination card control in this two-center study. Based on the recommendations of the Robert Koch Institute, we assessed the completeness of the vaccination status of the examined vaccinations. Furthermore, a comparative analysis of patients with complete/incomplete or correctly/wrongly self-reported vaccination status was performed. In the cohort analyzed, the vaccination coverage was 79.5% for tetanus, 79.2% for diphtheria, 74.8% for pertussis and 84.8% for poliomyelitis. The assumed vaccination status was higher for tetanus (86.5%) and lower for diphtheria (69.4%), pertussis (61.2%) and poliomyelitis (75.9%). Patients who were unvaccinated or only partially vaccinated against tetanus had received vaccination advice from a physician less often in the past year (13.4 vs. 36.9%, *p* < 0.001) and had no one to check the vaccination card more often (35.8 vs. 12.3%, *p* < 0.001). High sensitivity (93.7%) and low specificity (30.3%) were determined regarding the validity of self-reported tetanus vaccination status. Patients with a correctly reported tetanus vaccination status were more likely to have their vaccination card checked by a physician than those who overestimated or underestimated their vaccination status (76.7 vs. 63.0/43.8%, *p* = 0.002). Similar findings were seen with regard to diphtheria, pertussis and poliomyelitis vaccination. Patients without a regular vaccination card control (17.1%) were more likely to be male (44.6 vs. 29.4%, *p* = 0.037), had fewer siblings on average (1.1 vs. 1.6, *p* = 0.016), dealt less frequently with the issue of vaccination in the past year (32.1 vs. 69.3%, *p* < 0.001) and more frequently had the wish to receive vaccination advice (48.2 vs. 34.4%, *p* = 0.030) than patients in whom the vaccination card was checked regularly by a physician. To minimize the risk of infection in MS patients, treating physicians should provide regular vaccination counseling and perform vaccination card controls, as these factors are associated with a higher vaccination coverage and a higher validity of self-reported vaccination statuses.

## 1. Introduction

Multiple sclerosis (MS) is a chronic inflammatory and neurodegenerative disease of the central nervous system [1,2] that affects more than 2.8 million people worldwide [3], with a higher incidence in women than in men [4]. Generally, the disease is diagnosed between the ages of 20 and 49 years [5,6]. In the development of MS, genetic and environmental factors play a crucial role [7,8,9,10]. The symptoms can be remarkably diverse: Initially, patients often complain about dysesthesias, paresis, or visual problems. As the disease progresses, fatigue, spasticity, bladder dysfunction, ataxia, pain, depression and cognitive dysfunction are among the most commonly reported symptoms [11]. Several courses of the disease have been distinguished [12]. The most common disease course at onset is relapsing-remitting MS (RRMS), in which neurological deficits occur during relapses and resolve completely or incompletely. Secondary progressive MS (SPMS), which can develop from RRMS in the course of the disease and the rarer primary progressive MS (PPMS) are characterized by a steadily progressive accumulation of disability over time. In PPMS, there are no relapses from the onset of the disease. Clinically isolated syndrome (CIS), refers to patients who are affected by an initial clinical event that is strongly suggestive of MS but does not yet meet the diagnostic criteria for MS [13].Current treatment strategies for MS focus on treating acute relapses, relieving symptoms and reducing disease activity. These disease-modifying therapies influence the course of MS by suppressing or modulating immune functions [14]. More than a dozen disease-modifying drugs (DMDs) are currently available for the treatment of MS [15,16]. A disadvantage of these immunomodulatory and immunosuppressive therapies is the increased risk of infection observed in MS patients [17]. It has long been discussed that infections play a role in the development of MS [18,19,20]. In addition, other studies have shown that infections may increase the risk of relapses in MS patients [17,21,22], with more sustained disability (as measured by the Expanded Disability Status Scale (EDSS) [23]) compared to infection-independent relapses [24]. Furthermore, the hospitalization rate and the mortality of MS patients affected by infections is increased compared to those without infections [25,26].

Due to the adverse effects of infections, special attention should be paid to the prevention of infections in MS patients, for instance through vaccinations [27]. While vaccinations are considered one of the most important successes in the history of medicine, with vaccines having saved countless lives by preventing potentially dangerous and life-limiting infectious diseases, the relation of MS and vaccination has been debated for decades. However, studies have not confirmed the concern that vaccinations may exacerbate MS or even be a trigger of the disease (with the exception of yellow fever vaccination, which may increase the relapse risk) [28,29]. Overall, the advantages of vaccinations outweigh the potential risks and before starting DMD therapy, completion of vaccination status is recommended in MS patients [30]. In addition to this special priority of vaccinations for MS patients, we live in a time in which vaccination hesitancy plays an increasingly important role in society [31,32].

We aimed to examine the vaccination status (completeness), the validity of self-reported vaccination status, vaccination knowledge and vaccination behavior of MS patients. Furthermore, our objective was to determine whether the vaccination status and the vaccination behavior of MS patients are associated with sociodemographic or clinical patient characteristics. To our knowledge, this study is the first to present data on the willingness of MS patients to get vaccinated and their interest in vaccinations. We investigated vaccination statuses based on the common vaccinations for tetanus, diphtheria, pertussis and poliomyelitis because these have been administered as a combination vaccination in Germany for many years [33]. They have been administered in this manner as it is recommended that tetanus, diphtheria and pertussis vaccinations are boosted every ten years [34]. The investigation of the completeness of these important vaccinations allows us to gain an insight into the vaccination awareness of MS patients.

## 2. Materials and Methods

### 2.1. Study Cohort

This cross-sectional multi-center study was conducted at the Department of Neurology (Section of Neuroimmunology) of the Rostock University Medical Center and at the Neurological Department of the Ecumenical Hainich Hospital in Mühlhausen (Germany). Data from a total of 327 patients (157 from Rostock and 170 from Mühlhausen) with a confirmed diagnosis of MS according to the revised McDonald criteria from 2017 [13] were collected in the years 2019 and 2020. The medical centers have special outpatient and inpatient wards for MS patients. The inpatients were either hospitalized to relieve disease activity or drug side effects or were admitted for emergency care and further examinations. In the outpatient wards, patients usually had medical appointments for routine checkups. The participation of inpatients and outpatients in this study was on a voluntary basis after we gave them spoken and written information and an appropriate time to consider the opportunity to participate in this study. After obtaining informed consent, patient data were collected on the basis of four sources of information: an anamnesis as well as patient records (including the vaccination card), a clinical examination and a structured patient interview. This study was approved by the ethics committee of the University of Rostock, the ethics committee of Thuringia and conducted according to the Declaration of Helsinki.

### 2.2. Data Acquisition

We collected sociodemographic, clinical-neurological and vaccination data of the patients. Sociodemographic data included sex, age, years in school (not including training or higher education), educational level, employment status, partnership status and place of residence (rural community—≤5000 residents, provincial town—5000–19,999 residents, medium-sized town—20,000–99,999 residents, city—≥100,000 residents) as well as numbers of children and siblings.

The clinical-neurological data included type of care (inpatient or outpatient), course of MS, disease duration (years), degree of disability (EDSS), number of comorbidities (defined according to the recommendations from the “International Workshop on Comorbidities in MS”) [35,36] and type of DMD used.

Vaccination data covered information on the patients’ attitudes regarding vaccinations, e.g., whether they received a vaccination consultation in the past year, whether they dealt with the issue of vaccinations in the past year, the wish for vaccination advice, and information on who regularly checks the patient’s vaccination card. Additionally, we checked the patients’ vaccination cards for childhood basic immunizations and adult booster immunizations to evaluate the completeness of immunizations against tetanus, pertussis, diphtheria and poliomyelitis. To consider the vaccination status as complete, we followed the German national recommendations by the Standing Committee on Vaccination (STIKO) of the Robert Koch Institute (RKI) [37] (Table 1). Tetanus, diphtheria and pertussis vaccination statuses were considered complete if, in addition to basic immunization, a booster vaccination had been administered within the last ten years. Poliomyelitis vaccination status was defined as complete if the patients received the full basic immunization. We also obtained patients’ opinions on the completeness of tetanus, diphtheria, pertussis and poliomyelitis vaccinations in order to compare them with the actual vaccination statuses according to their vaccination cards.

### 2.3. Statistics

We analyzed the data of all patients whose vaccination cards indicated that they had either a complete or an incomplete vaccination status against tetanus, diphtheria, pertussis or poliomyelitis. Patients with an incomplete vaccination status were either unvaccinated or did not receive all necessary vaccinations. Missing data resulted when information was available for a patient for one of these vaccinations but not for another. Therefore, for pertussis and poliomyelitis vaccination, only 317 and 303 of the 327 patients were evaluated, respectively.

The statistical analysis was performed using PASW Statistics 27 (IBM). The data were tested for normal distribution using Kolmogorov–Smirnov tests. Means and standard deviations or medians and ranges are presented for metric data, whereas counts and percentages are reported for categorical data. Differences between patient groups were examined using one-way analysis of variance (ANOVA), chi-squared tests, Fisher’s exact tests, Kruskal–Wallis tests, Mann–Whitney U tests, McNemar’s test and two-tailed Student’s *t*-tests as appropriate. *p*-values < 0.05 indicated statistically significant differences. The false discovery rate (FDR) was applied to consider alpha error accumulation due to multiple testing [38]. Differences that were significant even after FDR correction were marked with an asterisk in the tables. The validity of self-reported vaccination status of fully and incompletely vaccinated patients was assessed using the measures sensitivity and specificity. Only those patients who reported their vaccination status as complete or incomplete were included in this calculation. Bar charts and boxplots were created using Microsoft Excel 2016 (Version 2203, Microsoft Corporation, Redmond, WA, USA).

## 3. Results

### 3.1. Sociodemographic and Clinical Data

The study included 327 MS patients with an average age of 47.3 ± 13.1 years and the proportion of women was 68.8%. The majority of patients lived in a partnership (75.2%), had ≥1 child (72.2%) and ≥1 sibling (90.2%). The duration of schooling was 10.6 ± 1.3 years and the majority of MS patients (60.6%) had completed training as a skilled worker. A subset of 41.0% of the patients were already retired due to illness or age. Clinical neurological examination revealed a median EDSS score of 3.0 (range: 0–8.0) with median disease duration of 10 years (range: 0–41 years). A total of 221 patients (67.6%) had a diagnosis of RRMS and almost 75% of the patients suffered from comorbidities in addition to MS. A subset of 78.9% of the patients were treated with a DMD at the time of the data acquisition (Table 2). The most commonly used DMDs were interferon-beta preparations (15.3% of the patients), glatiramer acetate (10.1%), fingolimod (8.0%), natalizumab (7.0%) and ocrelizumab (6.7%).

### 3.2. Vaccination Survey Data

Over 35% of the patients (*n* = 116) wished to receive vaccination advice from a physician and 67.9% (*n* = 222) of the patients did not receive a consultation on vaccination within the past year. Considering the awareness of vaccinations, 37.0% (*n* = 121) of the patients stated that they had last dealt with the topic more than a year ago. A subset of 17.1% (*n* = 56) reported not having their vaccination status checked by anyone, while 72.8% (*n* = 238) of the patients reported that their vaccination cards were routinely checked by their family doctor or by a neurologist and 10.1% (*n* = 33) indicated that other people checked their immunization records. By checking the patients’ vaccination cards, we found a complete vaccination status for tetanus in 79.5%, for diphtheria in 79.2%, for pertussis in 74.8% and for poliomyelitis in 84.8% of the patients. We compared these documented immunization statuses with the patients’ self-reported vaccination statuses. The assumed vaccination completeness was significantly higher for tetanus (*p* = 0.008) and significantly lower for the other three vaccinations (*p* ≤ 0.005) than the real vaccination status (Figure 1a). The proportion of patients who correctly assessed their own vaccination status was highest for tetanus (78.6%) and lowest for pertussis (56.5%). The proportion of patients who reported not knowing their vaccination status was 2.4% for tetanus and approximately 20% for the other vaccinations (Figure 1b).

### 3.3. Comparison between Patients with Complete and Incomplete Vaccination Status

We divided the patient cohort into patients with a complete vaccination status (PwCV) and patients with an incomplete vaccination status (PwIV) for each type of vaccination and compared the patient groups with regard to differences in sociodemographic, clinical, and vaccination data. The comparisons revealed no significant differences in terms of age and sex for all vaccinations (Table 3), although there was a consistent tendency that males were overrepresented among PwiV. PwIV for tetanus and diphtheria had significantly fewer siblings than those who were fully vaccinated (1.2 vs. 1.6; *p* ≤ 0.008). PwCV for poliomyelitis were found to be more likely to live in rural areas and cities compared to PwIV (*p* = 0.010). The groups did not differ with respect to other sociodemographic factors. Analysis of the clinical data also revealed no significant differences between the patient groups except for poliomyelitis vaccination: PwIV for poliomyelitis (*n* = 46) were significantly more likely to have SPMS than PwCV (28.3 vs. 16.3%; *p* = 0.013). For the vaccinations that need to be boosted every 10 years (tetanus, diphtheria and pertussis), differences between PwCV and PwIV were seen in vaccination awareness: significantly more PwIV did not receive vaccination counseling by a physician in the past year (*p* ≤ 0.001) (Figure 2a) and did not deal with the issue of vaccinations during the past year (Figure 2b). However, no significant differences could be elicited in the desire for vaccination advice between the groups (*p* ≥ 0.349). The proportion of patients without any independent vaccination card check was significantly higher in those patients that were incompletely immunized against tetanus, diphtheria and pertussis (*p* < 0.001) (Figure 2c). A similar but non-significant difference between PwCV and PwIV was found for poliomyelitis vaccination (*p* = 0.058). There were also significant differences between PwCV and PwIV in the correctness of self-reported vaccination statuses. For all vaccinations, we found high sensitivity and low specificity in self-reported vaccination status when compared to the documented vaccination status (*p* ≤ 0.014). Less than half of the PwIV correctly reported their vaccination status as incomplete for each of the four vaccinations (Figure 2d).

### 3.4. Comparison between Patients with Correct and Incorrect Vaccination Self-Assessment

Next, we examined patients with correctly and incorrectly self-reported vaccination status for differences in sociodemographic, clinical and vaccination data (Figure 3 and Appendix A). Misreported vaccination status was subdivided into overestimated and underestimated vaccination status. For tetanus, there was a higher proportion of male patients among the incorrectly reporting group. However, this difference was not statistically significant (*p* = 0.119). Significant differences were repeatedly found for education, disease duration, the date when the patients last dealt with the issue of vaccinations and vaccination card contact person (Appendix A). Underestimating patients (with respect to diphtheria and pertussis immunization) were in school for fewer years (*p* ≤ 0.006) than overestimating and correctly reporting patients. The disease duration was significantly longer in patients who correctly reported their vaccination status for tetanus and diphtheria than in those who reported it incorrectly (*p* ≤ 0.026). This difference was also seen for the self-assessment regarding vaccination against pertussis and poliomyelitis but did not reach the significance level. Significantly more patients with correctly reported vaccination status for tetanus, diphtheria and poliomyelitis had dealt with the topic of vaccinations within the past year compared to patients who reported their vaccination status incorrectly or as unknown (*p* ≤ 0.024). For all four vaccinations, a significantly higher proportion of correctly reporting patients had their vaccination card checked by their family doctor/neurologist, whereas the proportion of patients who did not have a contact person on the subject of vaccinations was significantly lower compared to incorrectly reporting patients (*p* ≤ 0.008).

### 3.5. Comparison of Patients with Different Persons Who Regularly Check Their Vaccination Cards

In the last part, the patients were stratified according to the persons who regularly check their vaccination card: physicians (family doctor/neurologist), other persons or nobody (Table 4). The patients for whom nobody checked the vaccination card were significantly more often men (44.6 vs. 29.4%; *p* = 0.037) and had significantly fewer siblings than those with a physician contact person (1.1 vs. 1.6; *p* = 0.016). Furthermore, the patient group “nobody controls the vaccination card” was younger (44.8 vs. 47.9 years), had fewer children on average (1.0 vs. 1.3) and had a lower median disease duration (8.5 vs. 10.0 years) than the patient group “vaccination card checked by a physician”. However, these differences were not statistically significant (*p* > 0.05).

Analyzing the vaccination survey data, the rate of patients who received a vaccination advice in the last year was more than twice as high for patients who had their vaccination card checked by their physician than for patients without an independent vaccination card check (38.7 vs. 12.5%; *p* < 0.001). In the group “nobody checked the vaccination card”, significantly fewer patients had dealt with the topic of vaccination within the last year compared to patients where a physician checked their vaccination card (32.1 vs. 69.3%; *p* < 0.001). On closer inspection, males without a vaccination card check showed as a proportion of 72.0%, the highest proportion of patients who had last dealt with the issue of vaccinations more than a year ago (Figure 4). Of note, significantly more patients without independent vaccination card control had a desire for a vaccination advice than patients with regular vaccination card check by a physician (48.2 vs. 34.4%, *p* = 0.030).

## 4. Discussion

Due to the high relevance of an appropriate vaccination status for MS patients, the aim of our study was to investigate the vaccination status, vaccination attitude and vaccination card knowledge of patients with MS. In those patients, indication-appropriate vaccinations can be considered largely safe as the benefits outweigh the disadvantages, and, therefore, they are usually recommended [29,30,39]. An exception is the use of certain live vaccines, which are contraindicated under immunosuppressive medication. Inactivated vaccines, on the other hand, can generally be used without an increased risk of adverse effects [40]. Infectious diseases pose a severe risk to patients with MS [17,26]. Disease-modifying therapies for MS, aside from interferon-beta and glatiramer acetate, make patients more susceptible to infections by suppressing or modulating normal immune response [41,42]. To minimize this risk, generally recommended vaccinations against vaccine-preventable diseases are of utmost importance. To be adequately protected, vaccination gaps should be closed, ideally before starting a DMD therapy [30]. In our analysis, we focused on tetanus, diphtheria, pertussis and poliomyelitis vaccines, which are all inactivated vaccines and are considered safe or probably safe for patients with MS according to recent studies [29]. In addition, it is recommended by the STIKO to have booster vaccinations for tetanus, diphtheria and pertussis every ten years, which is why these vaccinations are relevant in the clinical routine of physicians treating MS patients [37]. Our study population was a representative MS patient cohort in terms of age, sex and disease course when compared with data from the German MS registry [6,43].

Considering the vaccination coverage of our patients, a complete vaccination status was found for tetanus in 79.5%, for diphtheria in 79.2%, for pertussis in 74.8% and for poliomyelitis in 84.8% of the patients. The finding that the proportion of fully vaccinated patients was lowest for pertussis compared with the other vaccinations is presumably explained by the fact that the recommendation to combine the next due tetanus-diphtheria vaccination with a pertussis vaccine was issued only in 2009 [44]. Prior to 2009, there was no recommendation for booster pertussis vaccination in adulthood. This underscores the observation that the use of combination vaccines is associated with improved coverage rates [45]. To our knowledge, there are no previous data on vaccination coverage rates for tetanus, diphtheria, pertussis and poliomyelitis in MS patients.

However, when compared to the general population in Germany, we observed markedly higher vaccination coverage rates than reported in the latest national evaluation by the federal government’s research institute for disease control and prevention (Robert Koch Institute). Among more than 56 million people for whom vaccination data were assessed, coverage rates were only 51.9% for tetanus, 53.3% for diphtheria and 41.9% for pertussis [34]. No current data are available for poliomyelitis. The most recent findings indicate a lifetime prevalence of at least one polio vaccination of 85.6% [46]. In the studies on nationwide vaccination coverage rates, the proportions of people with complete vaccination status were between 5 and 25% higher in eastern German states than in western German federal states [34,46,47,48]. Studies that have examined vaccinations in patients with other autoimmune diseases report relatively low vaccination completion rates for tetanus, diphtheria, pertussis and poliomyelitis [49,50,51,52,53,54]. For instance, in the study by Chehab et al., of the 579 lupus patients examined, 65.8% had a complete vaccination status for tetanus [49]. The higher vaccination coverage rates in our MS patients as compared with nationwide data and the vaccination coverage rates of patients with other autoimmune diseases may be due to more regular clinical visits and thus more frequent vaccination card checks in MS patient care. In addition, our study was conducted at centers in two eastern German federal states, where a historically higher willingness to vaccinate and a higher acceptance of vaccination recommendations favor a more complete vaccination status [46,55]. Besides these facts, years of discussion about the potential risks and benefits of vaccinations in MS may have increased the attention on patients’ vaccination status by treating physicians [28,30,56,57,58]. In addition to the vaccinations examined in this study, further research on other vaccinations is needed to provide a more complete view of the vaccination status and the willingness of MS patients to get vaccinated. The annual influenza vaccination would be of great interest in this regard, as it implies more regular vaccination status tracking by treating physicians. Furthermore, the vaccination status for live vaccines such as the varicella vaccine should be investigated, as these are more controversial in the context of immunomodulatory treatments than the safer inactivated vaccines [40]. However, for live vaccines, completeness is often achieved through childhood baseline immunization before the patients develop MS and therefore does not provide information on the vaccination awareness of MS patients [37].

The comparison of PwCV with PwIV revealed that patients who were fully vaccinated against tetanus and diphtheria had significantly more siblings on average than those who were unvaccinated or partially vaccinated (1.6 vs. 1.2; *p* ≤ 0.008). Inconsistent findings have been reported on this in the literature. For instance, it has been found that children with a higher number of siblings more often do not have complete vaccination status against measles [59]. Differences in study design (e.g., children vs. adults, type of vaccination studied, etc.) may explain such discrepancies. Other associations that were described in the literature, such as increased tetanus vaccination rates in men [60] and in patients with higher levels of education [61], were not reflected in our results. In clinical practice, it must be considered that there may be a reduced immune response following vaccination in patients receiving a DMD for MS. Antibody titers may be determined after vaccination, since studies have shown reduced antibody responses under therapy with DMDs, with the exception of beta-interferons [62,63]. Our findings, however, demonstrate that vaccination coverage is mostly influenced by vaccination attitudes. We found that vaccination consultations and vaccination card checks by physicians were clearly related to vaccination completeness. This confirmed results from previous studies show that patients with vaccination counseling or vaccination card control by a physician were significantly more likely to be fully vaccinated and that the family doctor/specialist is typically the most important advisor on vaccinations [47,53]. The relevance of vaccination advice for MS patients has been demonstrated in our study and should prompt physicians to perform regular vaccination counseling and vaccination card checks in order to reduce the risk of infection for MS patients. Continuous vaccination monitoring in the form of an electronic vaccination card could be helpful in the future to detect PwIV and make them an offer of vaccination [64]. Some PwIV may not be adequately informed about vaccination recommendations, such as the need for booster vaccinations for tetanus, diphtheria and pertussis [65]. On the other hand, some PwIV may reject vaccinations in general due to their attitudes or personal beliefs, and therefore do not want to be vaccinated and are not interested in their own vaccination status [66]. These considerations need to be taken into account in further studies looking at the vaccination attitudes, knowledge of vaccination recommendations and personalities of MS patients.

Concerning the validity of patients’ self-reported vaccination status, a high sensitivity and a low specificity were found for all vaccinations, implying that PwIV are poorly aware of their own vaccination status and that they are more likely to think that they are fully vaccinated even though they are not. These results are consistent with those in the current literature on tetanus vaccination self-reports [67,68]. A high sensitivity of tetanus vaccination self-reports may reflect a high awareness of the tetanus vaccination [69]. Tetanus is caused by bacteria that enter the body through contaminated wounds. Since people are frequently confronted with this danger, for example when they are injured while gardening, they presumably deal with their tetanus vaccination status more frequently or are more aware of the tetanus vaccination [70]. In contrast to our results are the findings from the study by Loulergue et al., who observed low sensitivity (<50%) and high specificity (70–95%) for tetanus, diphtheria, pertussis and polio vaccination self-reports [71] but this study was conducted in French health care students with an average age of 23 years. In the patient interviews, we got the impression that in the context of combination vaccinations containing tetanus, some patients were not sufficiently informed about the additionally included vaccines besides tetanus.

In the analyses of patients with correctly vs. incorrectly self-reported vaccination status, the main differences were found in education, disease duration and vaccination behavior. A low educational level was a predictor for underestimating vaccination completeness. This fact was seen for diphtheria as well as for pertussis but not for tetanus, possibly because patients with lower levels of education were not adequately informed about the inclusion of diphtheria and pertussis vaccine in the decennial booster with tetanus. In general, low school qualifications have been shown to be associated with a low health status and a low health literacy [72]. Patients with correct reports had a longer disease duration than those who were incorrect. The longer disease duration could have led patients to pay more attention to their state of health and their vaccination status over time. Interest in vaccinations, on the one hand, and implementation of a vaccination card control, on the other hand, emerged as parameters that distinguished our patients with correctly and incorrectly self-reported vaccination status. Regular physician vaccination advice and stronger interest in vaccinations increased the validity of self-reported immunization status in our patients. A possible implication of these results is that physicians should conduct regular vaccination card controls and educate their MS patients about their vaccination status and the vaccines included in combination vaccinations. In this context, special attention should be paid to MS patients with low levels of education and short disease durations.

In our patient cohort, the most frequently mentioned control authority of the vaccination card was the family doctor/neurologist with 72.8%. In contrast, approximately 17% of our patients stated that they did not have their vaccination card checked by anyone. These results are consistent with the literature on patients with other autoimmune diseases, where the family doctor/specialist was stated as the person checking the immunization record by 69% of the patients [54] and 24% of the patients did not have regular vaccination card controls [49]. In the present study, as well as in the literature, vaccination card control was associated with patients with a more complete vaccination status and a better knowledge of their own vaccination status [53]. This raises the question of what distinguishes patients with and without vaccination card control. In our data, men more often did not have a contact person for an independent vaccination card check, which may reflect the generally higher utilization of medical services by women [73]. Of those patients without vaccination card control, a lower proportion dealt with the subject of vaccinations within the past year than patients with control by a physician (32.1 vs. 69.3%, *p* < 0.001). However, among patients without a check, a larger proportion expressed a desire for vaccination counseling compared to patients with control (48.2 vs. 34.4%, *p* = 0.030), emphasizing that some patients are clearly interested in discussing their own vaccination status with a physician. In particular, for male patients with few siblings, more attention should be paid to offering them regular vaccination card control, as they are less likely to be completely vaccinated.

A potential limitation of our study is that the patient recruitment ran through June 2020, which may have caused the results to be skewed by media attention of the SARS-CoV-2 pandemic beginning in March 2020. Due to the potential risk of SARS-CoV-2 infection, more patients may have been concerned with vaccinations and their vaccination status. A further limitation is that our patients came almost exclusively from the former east Germany, and hence no national data were collected, which could have led to some bias. While we included only patients who are treated in a hospital, there could be differences with patients who are treated exclusively by neurologists in private practices. Furthermore, we only collected cross-sectional data and no longitudinal data of the patients. Last but not least, since only four selected vaccinations were examined by the authors, the present study cannot provide an overall view of the vaccination status of MS patients, according to the RKI recommendations on standard, indications, and travel vaccinations [37]. Nonetheless, to the best of our knowledge, this study was the first to collect data on vaccination rates, vaccination knowledge and vaccination behavior in MS patients, which may serve as the basis for further studies that are clearly needed to further improve the monitoring of vaccination rates in MS patients in the future.

## 5. Conclusions

In summary, vaccination rates against tetanus, diphtheria, pertussis, and poliomyelitis within our MS patient cohort were higher than for the average German population and patients with other autoimmune diseases. However, as vaccinations are of particular importance for MS patients, treating physicians, especially family doctors and neurologists, should offer regular vaccination consultations to their patients and check vaccination cards, as both vaccination coverage and patients’ knowledge of their own vaccination status are significantly influenced by these factors. To increase the congruence of assumed and real vaccination status, patients should be informed of all the vaccines contained in a combination vaccination. Furthermore, special attention should be paid to men and patients with few siblings, since these patients more often did not have their vaccination cards checked by an independent person. To prove or reject causality, these factors should be investigated in long-term studies. Continuous vaccination monitoring, for example in the form of individual electronic vaccination cards, could help physicians keep better track of their patients’ vaccination status. Further research is needed to explore reasons for incomplete vaccination status, poor vaccination knowledge and vaccination refusal in patients with MS.

## Figures and Tables

**Figure 1 jpm-12-00677-f001:**
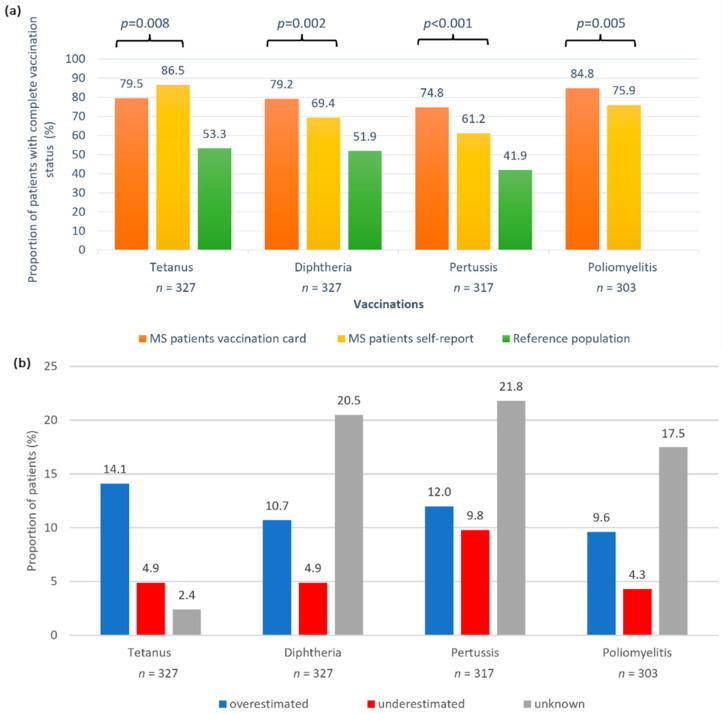
Concordance of self-reported vaccination status and vaccination card information in patients with MS. (**a**) There was a significant difference in immunization status as documented in the vaccination cards (orange bars) and as assumed by the patients (yellow bars). More patients with MS thought they were fully vaccinated against tetanus than was actually the case (*p* < 0.008 ^McNemar^). In contrast, for the other vaccinations, the assumed vaccination status was lower than the documented vaccination rates: more patients were actually fully vaccinated than the self-reports indicated (*p* ≤ 0.005 ^McNemar^). Vaccination rates were markedly higher than for the general population in Germany (no data on polio vaccines were available) [34]. (**b**) More patients overestimated vaccination completeness than underestimated it. The proportion of patients who reported not knowing their vaccination status was 2.4% for tetanus and roughly 20% for the other vaccinations under consideration. ^McNemar^—McNemar’s test; MS—multiple sclerosis; *n*—number of patients; *p*—*p*-value.

**Figure 2 jpm-12-00677-f002:**
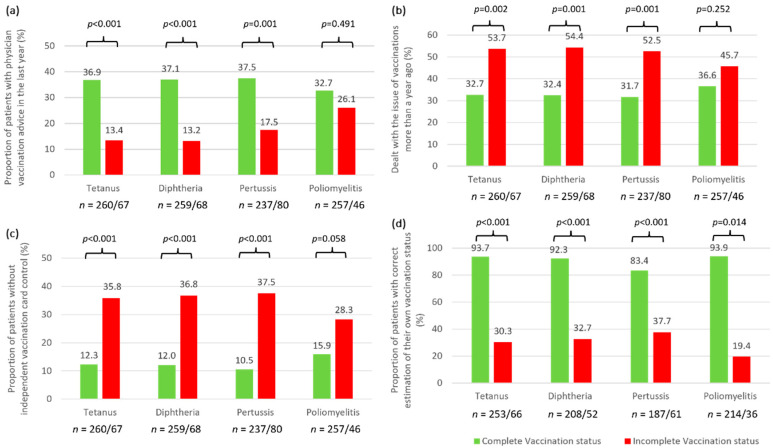
Comparison of MS patients with complete vs. incomplete vaccination status with regard to vaccination advice, interest, control and knowledge. (**a**) The proportion of patients with physician counseling on vaccination within the past year was generally lower among PwIV than among PwCV. (**b**) A higher proportion of PwIV did not deal with the issue of vaccinations during the past year. (**c**) Comparing the two groups in terms of general vaccination card control, the proportion of patients whose vaccination card was not regularly checked by anyone was markedly higher among PwIV. These differences were significant for all vaccinations (*p* < 0.001 ^Fi^), except for polio vaccination. (**d**) Concerning patients’ knowledge of their own vaccination status, the proportion of patients who correctly estimated it was significantly lower for PwIV compared to PwCV (*p* ≤ 0.014 ^Fi^). ^Fi^—Fisher’s exact test; *n*—number of patients; *p*—*p*-value; PwCV—patients with complete vaccination status; PwIV—patients with incomplete vaccination status.

**Figure 3 jpm-12-00677-f003:**
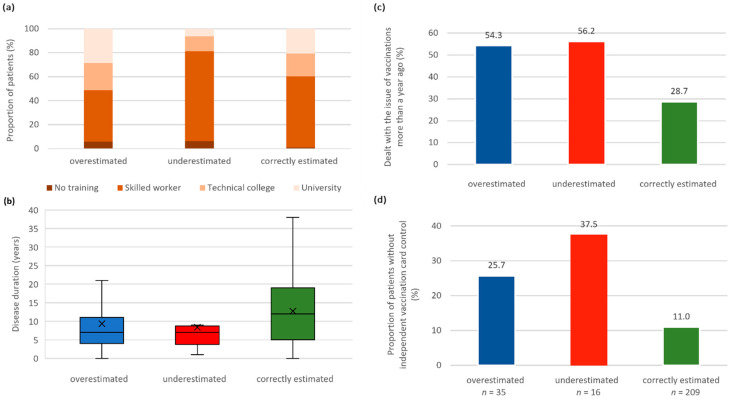
Assessment of diphtheria vaccination status by MS patients in relation to education, disease duration, vaccination interest, and vaccination control. Patients who underestimated their vaccination status were significantly more likely to be educated as skilled workers and less likely to have a university degree (*p* = 0.004 ^Chi^) (**a**). Patients who correctly estimated their vaccination status had a significantly longer disease duration (*p* = 0.004 ^Chi^) (**b**), were less likely to not address the issue of vaccinations for more than a year (*p* = 0.001 ^Chi^) (**c**) and were less likely to not have a contact person for vaccination card control (*p* = 0.005 ^Chi^) (**d**). ^Chi^—chi-squared test; *n*—number of patients; *p*—*p*-value.

**Figure 4 jpm-12-00677-f004:**
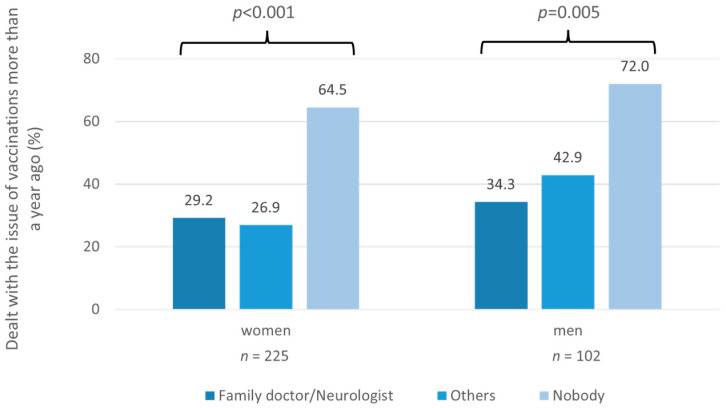
Vaccination interest of MS patients stratified by sex and by persons who regularly checked their vaccination card. Women and men were separately examined for an association between their interest in vaccinations and the person checking their vaccination card. Of the men who did not have their vaccination card checked by anyone, about 70% had not dealt with the issue of vaccinations within the past year (chi-squared test: *p* = 0.005). *n*—number of patients; *p*—*p*-value.

**Table 1 jpm-12-00677-t001:** German national recommendations by the Standing Committee on Vaccination (STIKO) of the Robert Koch Institute (RKI) for tetanus, diphtheria, pertussis and poliomyelitis vaccination [37].

Vaccination	RKI Recommendations 2020/21		
	Childhood Basic Immunization	Childhood Booster Immunization	Adult Immunization
Tetanus	Three vaccinations at 2, 4 and 11 months of age	At 5–6 years and 9–16 years of age	All persons in case of missing or incomplete basic immunization or if the last basic immunization or the last booster vaccination is longer than 10 years ago
Diphtheria	Three vaccinations at 2, 4 and 11 months of age	At 5–6 years and 9–16 years of age	All persons in case of missing or incomplete basic immunization or if the last basic immunization or the last booster vaccination is longer than 10 years ago
Pertussis	Three vaccinations at 2, 4 and 11 months of age	At 5–6 years and 9–16 years of age	It is recommended that the next Td vaccinations given as a single Tdap combination vaccination
Poliomyelitis	Three vaccinations at 2, 4 and 11 months of age	Between the ages of 9–16 years	All persons with missing or incomplete basic immunization and all persons without a booster vaccination

RKI—Robert Koch Institute; Td—tetanus-diphtheria vaccine; Tdap—tetanus-diphtheria-pertussis vaccine.

**Table 2 jpm-12-00677-t002:** Sociodemographic and clinical data of the examined patients with multiple sclerosis (*n* = 327).

	*n* (%)	Range	Mean (SD)	Median
**Sex**				
Female	225 (68.8)			
Male	102 (31.2)			
**Age (years)**		19–80	47.3 (13.1)	49.0
**Place of residence**				
Rural area	133 (40.7)			
Small town	49 (15.0)			
Medium-sized town	61 (18.7)			
City	84 (25.7)			
**Partnership**				
Single	61 (18.7)			
In a relationship	65 (19.9)			
Married	181 (55.4)			
Divorced	12 (3.7)			
Widowed	8 (2.4)			
**Number of children**		0–4	1.2 (0.9)	1.0
0	91 (27.8)			
1	88 (26.9)			
2	130 (39.8)			
3	15 (4.6)			
4	3 (0.9)			
**Number of siblings**		0–9	1.5 (1.1)	1.0
0	32 (9.8)			
1	169 (51.7)			
≥2	126 (38.5)			
**School years**		8–18	10.6 (1.3)	10.0
**Educational level**				
No training	11 (3.4)			
Skilled worker	198 (60.6)			
Technical college	59 (18.0)			
University	59 (18.0)			
**Employment status**				
Training/student	5 (1.5)			
Fulltime employed	94 (28.7)			
Part-time employed	74 (22.6)			
Unemployed	13 (4.0)			
Disability pension	105 (32.1)			
Retirement pension	29 (8.9)			
Other	7 (2.1)			
**Patient care**				
Outpatients	258 (78.9)			
Inpatients	69 (21.1)			
**Course of disease**				
CIS	16 (4.9)			
RRMS	221 (67.6)			
SPMS	63 (19.3)			
PPMS	27 (8.3)			
**Disease duration (years)**		0–41	11.6 (8.6)	10.0
**EDSS score**		0–8.0	3.3 (2.1)	3.0
**DMD treatment**				
Yes	258 (78.9)			
No	69 (21.1)			
**Comorbidities**				
Yes	244 (74.6)			
No	83 (25.4)			

CIS—clinically isolated syndrome; DMD—disease-modifying drug; EDSS—Expanded Disability Status Scale; MS—multiple sclerosis; *n*—number of patients; PPMS—primary progressive MS; RRMS—relapsing-remitting MS; SD—standard deviation; SPMS—secondary progressive MS.

**Table 3 jpm-12-00677-t003:** Comparison of MS patients with complete vs. incomplete vaccination status with regard to sociodemographic and clinical data.

	Tetanus (*n* = 327)		Diphtheria (*n* = 327)		Pertussis (*n* = 317)		Polio (*n* = 303)	
	Complete vs. Incomplete	*p*-Value	Complete vs. Incomplete	*p*-Value	Complete vs. Incomplete	*p*-Value	Complete vs. Incomplete	*p*-Value
*n*	260 vs. 67		259 vs. 68		237 vs. 80		257 vs. 46	
**Sex ^c^**		0.077 ^Fi^		0.056 ^Fi^		0.069 ^Fi^		0.302 ^Fi^
Female	185 (71.2) vs. 40 (59.7)		185 (71.4) vs. 40 (58.8)		170 (71.7) vs. 48 (60.0)		179 (69.7) vs. 28 (60.9)	
Male	75 (28.8) vs. 27 (40.3)		74 (28.6) vs. 28 (41.2)		67 (28.3) vs. 32 (40.0)		78 (30.3) vs. 18 (39.1)	
**Age (years) ^a^**	47.7 (12.9) vs. 45.9 (13.8)	0.325 ^t^	47.7 (12.9) vs. 45.9 (13.7)	0.306 ^t^	47.8 (12.9) vs. 45.2 (13.5)	0.127 ^t^	46.7 (12.7) vs. 49.2 (14.3)	0.213 ^t^
**Place of residence ^c^**		0.284 ^Chi^		0.322 ^Chi^		0.135 ^Chi^		**0.010** ^Chi^
Rural area	109 (41.9) vs. 24 (35.8)		108 (41.7) vs. 25 (36.8)		102 (43.0) vs. 27 (33.8)		112 (43.6) vs. 15 (32.6)	
Small town	34 (13.1) vs. 15 (22.4)		34 (13.1) vs. 15 (22.0)		29 (12.2) vs. 18 (22.5)		33 (12.8) vs. 13 (28.3)	
Medium-sized town	50 (19.2) vs. 11 (16.4)		50 (19.3) vs. 11 (16.2)		44 (18.6) vs. 14 (17.5)		40 (15.6) vs. 11 (23.9)	
City	67 (25.8) vs. 17 (25.4)		67 (25.9) vs. 17 (25.0)		62 (26.2) vs. 21 (26.2)		72 (28.0) vs. 7 (15.2)	
**Partnership ^c^**		0.872 ^Chi^		0.773 ^Chi^		0.898 ^Chi^		0.143 ^Chi^
Single	48 (18.5) vs. 13 (19.4)		48 (18.5) vs. 13 (19.1)		43 (18.1) vs. 17 (21.3)		48 (18.7) vs. 11 (23.9)	
In a relationship	49 (18.8) vs. 16 (23.9)		48 (18.5) vs. 17 (25.0)		46 (19.4) vs. 18 (22.5)		56 (21.8) vs. 6 (13.0)	
Married	147 (56.5) vs. 34 (50.7)		147 (56.8) vs. 34 (50.0)		134 (56.5) vs. 40 (50.0)		140 (54.5) vs. 24 (52.2)	
Divorced	10 (3.8) vs. 2 (3.0)		10 (3.9) vs. 2 (2.9)		8 (3.4) vs. 3 (3.7)		6 (2.3) vs. 4 (8.7)	
Widowed	6 (2.3) vs. 2 (3.0)		6 (2.3) vs. 2 (2.9)		6 (2.5) vs. 2 (2.5)		7 (2.7) vs. 1 (2.2)	
**Number of children ^a^**	1.3 (0.9) vs. 1.2 (0.9)	0.479 ^t^	1.3 (0.9) vs. 1.2 (0.9)	0.542 ^t^	1.3 (0.9) vs. 1.1 (0.9)	0.248 ^t^	1.2 (0.9) vs. 1.2 (1.0)	0.644 ^t^
**Number of siblings ^a^**	1.6 (1.2) vs. 1.2 (0.8)	**0.008 *** ^,t^	1.6 (1.2) vs. 1.2 (0.8)	**0.007 *** ^,t^	1.6 (1.1) vs. 1.3 (1.1)	0.083 ^t^	1.4 (1.0) vs. 1.7 (1.1)	0.159 ^t^
**School years ^a^**	10.6 (1.3) vs. 10.6 (1.2)	0.959 ^t^	10.6 (1.3) vs. 10.6 (1.2)	0.992 ^t^	10.6 (1.3) vs. 10.7 (1.3)	0.270 ^t^	10.7 (1.3) vs. 10.3 (1.1)	0.090 ^t^
**Educational level ^c^**		0.826 ^Chi^		0.813 ^Chi^		0.131 ^Chi^		0.160 ^Chi^
No training	8 (3.1) vs. 3 (4.5)		8 (3.1) vs. 3 (4.4)		7 (2.9) vs. 4 (5.0)		5 (2.0) vs. 3 (6.5)	
Skilled worker	160 (61.5) vs. 38 (56.7)		160 (61.8) vs. 38 (55.9)		148 (62.5) vs. 40 (50.0)		153 (59.5) vs. 28 (60.9)	
Technical college	47 (18.1) vs. 12 (17.9)		46 (17.8) vs. 13 (19.1)		44 (18.6) vs. 15 (18.7)		47 (18.3) vs. 10 (21.7)	
University	45 (17.3) vs. 14 (20.9)		45 (17.4) vs. 14 (25.6)		38 (16.0) vs. 21 (26.3)		52 (20.2) vs. 5 (10.9)	
**Employment status ^c^**		0.796 ^Chi^		0.782 ^Chi^		0.911 ^Chi^		0.646 ^Chi^
Training/student	4 (1.5) vs. 1 (1.5)		4 (1.5) vs. 1 (1.5)		3 (1.3) vs. 2 (2.5)		4 (1.6) vs. 1 (2.2)	
Full time employed	74 (28.5) vs. 20 (29.8)		73 (28.2) vs. 21 (30.9)		70 (29.5) vs. 23 (28.7)		82 (31.9) vs. 9 (19.5)	
Part timer employed	59 (22.7) vs. 15 (22.4)		59 (22.8) vs. 15 (22.1)		55 (23.2) vs. 18 (22.5)		59 (22.9) vs. 11 (23.9)	
Unemployed	8 (3.1) vs. 5 (7.5)		8 (3.1) vs. 5 (7.3)		8 (3.4) vs. 5 (6.3)		11 (4.3) vs. 1 (2.2)	
Disability pension	85 (32.7) vs. 20 (29.8)		85 (32.8) vs. 20 (29.4)		74 (31.2) vs. 24 (30.0)		74 (28.8) vs. 18 (39.1)	
Retirement pension	24 (9.2) vs. 5 (7.5)		24 (9.3) vs. 5 (7.3)		22 (9.3) vs. 6 (7.5)		21 (8.2) vs. 5 (10.9)	
Other	6 (2.3) vs. 1 (1.5)		6 (2.3) vs. 1 (1.5)		5 (2.1) vs. 2 (2.5)		6 (2.3) vs. 1 (2.2)	
**Patient care ^c^**		0.507 ^Fi^		0.617 ^Fi^		0.160 ^Fi^		0.243 ^Fi^
Outpatients	207 (79.6) vs. 51 (76.1)		206 (79.5) vs. 52 (76.5)		190 (80.2) vs. 58 (72.5)		205 (79.8) vs. 33 (71.7)	
Inpatients	53 (20.4) vs. 16 (23.9)		53 (20.5) vs. 16 (23.5)		47 (19.8) vs. 22 (37.5)		52 (20.2) vs. 13 (28.3)	
**Course of disease ^c^**		0.588 ^Chi^		0.311 ^Chi^		0.422 ^Chi^		**0.013** ^Chi^
CIS	11 (4.2) vs. 5 (7.5)		10 (3.9) vs. 6 (8.8)		10 (4.2) vs. 6 (7.5)		12 (4.7) vs. 4 (8.7)	
RRMS	179 (68.8) vs. 42 (62.7)		179 (69.1) vs. 42 (61.8)		167 (70.5) vs. 49 (61.3)		185 (72.0) vs. 22 (47.8)	
SPMS	48 (18.5) vs. 15 (22.3)		48 (18.5) vs. 15 (22.1)		41 (17.3) vs. 17 (21.2)		42 (16.3) vs. 13 (28.3)	
PPMS	22 (8.5) vs. 5 (7.5)		22 (8.5) vs. 5 (7.3)		19 (8.0) vs. 8 (10.0)		18 (7.0) vs. 7 (15.2)	
**Disease duration (years) ^b^**	10.0 vs. 8.0	0.096 ^U^	12.0 vs. 9.8	0.059 ^U^	10.0 vs. 7.5	0.050 ^U^	10.0 vs. 9.0	0.468 ^U^
**EDSS score ^b^**	3.0 vs. 2.5	0.550 ^U^	3.0 vs. 2.5	0.416 ^U^	3.0 vs. 3.0	0.641 ^U^	3.0 vs. 3.5	0.331 ^U^
**DMD treatment ^c^**		1.000 ^Fi^		0.868 ^Fi^		0.875 ^Fi^		0.847 ^Fi^
Yes	205 (78.9) vs. 53 (79.1)		205 (79.1) vs. 53 (77.9)		185 (78.1) vs. 64 (80.0)		200 (77.8) vs. 36 (78.3)	
No	55 (21.1) vs. 14 (20.9)		54 (20.9) vs. 15 (22.1)		52 (21.9) vs. 16 (20.0)		57 (22.2) vs. 10 (21.7)	
**Comorbidities ^c^**		0.532 ^Fi^		0.639 ^Fi^		0.140 ^Fi^		0.720 ^Fi^
Yes	196 (75.4) vs. 48 (71.6)		195 (75.3) vs. 49 (72.1)		181 (76.4) vs. 54 (67.5)		187 (72.8) vs. 35 (76.1)	
No	64 (24.6) vs. 19 (28.4)		64 (24.7) vs. 19 (27.9)		56 (23.6) vs. 26 (32.5)		70 (27.2) vs. 11 (23.9)	

CIS—clinically isolated syndrome; DMD—disease-modifying drug; EDSS—Expanded Disability Status Scale; MS—multiple sclerosis; *n*—number of patients; PPMS—primary progressive MS; RRMS—relapsing-remitting MS; SD—standard deviation; SPMS—secondary progressive MS. ^a^ mean value (standard deviation); ^b^ median; ^c^ number of patients (%); ^Chi^ chi-squared test; ^Fi^ Fisher’s exact test; ^t^ two-sample two-tailed Student’s *t*-test; ^U^ Mann–Whitney U test; * significant after FDR correction.

**Table 4 jpm-12-00677-t004:** Comparison of sociodemographic and clinical data between MS patients with different persons checking their vaccination card.

	Nobody	Family Doctor/ Neurologist	Others	*p*-Value
*n*	56 (17.1%)	238 (72.8%)	33 (10.1%)	
**Sex ^c^**				**0.037 ^Chi^**
Female	31 (55.4)	168 (70.6)	26 (78.8)	
Male	25 (44.6)	70 (29.4)	7 (21.2)	
**Age (years) ^a^**	44.8 (13.3)	47.9 (13.2)	48.0 (11.5)	0.270 ^ANOVA^
**Place of residence ^c^**				0.136 ^Chi^
Rural area	25 (44.6)	91 (38.2)	17 (51.5)	
Small town	6 (10.7)	41 (17.2)	2 (6.1)	
Medium-sized town	13 (23.2)	39 (16.4)	9 (27.3)	
City	12 (21.4)	67 (28.1)	5 (15.1)	
**Partnership ^c^**				0.625 ^Chi^
Single	14 (25.0)	41 (17.2)	6 (18.2)	
In a relationship	12 (21.4)	44 (18.5)	9 (27.3)	
Married	26 (46.4)	137 (57.6)	18 (54.5)	
Divorced	2 (3.6)	10 (4.2)	0 (0.0)	
Widowed	2 (3.6)	6 (2.5)	0 (0.0)	
**Number of children ^a^**	1.0 (0.9)	1.3 (0.9)	1.4 (1.1)	0.121 ^ANOVA^
**Number of siblings ^a^**	1.1 (0.8)	1.6 (1.2)	1.6 (0.9)	**0.016 ^ANOVA^**
**School years ^a^**	10.3 (1.1)	10.6 (1.3)	10.8 (1.3)	0.151 ^ANOVA^
**Educational level ^c^**				0.126 ^Chi^
No training	2 (3.6)	8 (3.4)	1 (3.0)	
Skilled worker	37 (66.1)	142 (59.7)	19 (57.6)	
Technical college	7 (12.5)	50 (21.0)	2 (6.1)	
University	10 (17.9)	38 (16.0)	11 (33.3)	
**Employment status ^c^**				0.797 ^Chi^
Training/student	2 (3.6)	3 (1.3)	0 (0.0)	
Fulltime employed	18 (32.1)	68 (28.6)	8 (24.2)	
Part-timer employed	9 (16.1)	54 (22.7)	11 (33.3)	
Unemployed	3 (5.4)	9 (3.8)	1 (3.0)	
Disability pensioned	20 (35.7)	75 (31.5)	10 (30.3)	
Retirement pensioned	3 (5.4)	23 (9.7)	3 (9.1)	
Other	1 (1.8)	6 (2.5)	0 (0.0)	
**Patient care ^c^**				0.616 ^Chi^
Outpatients	45 (80.4)	185 (77.7)	28 (84.8)	
Inpatients	11 (19.6)	53 (22.3)	5 (15.2)	
**Course of disease**				0.833 ^Chi^
CIS	4 (7.1)	11 (4.6)	1 (3.0)	
RRMS	36 (64.3)	161 (67.6)	24 (72.7)	
SPMS	12 (21.4)	44 (18.5)	7 (21.2)	
PPMS	4 (7.1)	22 (9.2)	1 (3.0)	
**Disease duration (years) ^b^**	8.5	10.0	12.0	0.216 ^H^
**EDSS score ^b^**	3.0	3.0	2.5	0.886 ^H^
**DMD treatment ^c^**				0.769 ^Chi^
Yes	42 (75.0)	189 (79.4)	26 (78.8)	
No	14 (25.0)	49 (20.6)	7 (21.2)	
**Comorbidities ^c^**				0.643 ^Chi^
Yes	39 (69.6)	180 (75.6)	25 (75.8)	
No	17 (30.4)	58 (24.4)	8 (24.2)	

CIS—clinically isolated syndrome; DMD—disease-modifying drug; EDSS—Expanded Disability Status Scale; MS—multiple sclerosis; *n*—number of patients; PPMS—primary progressive MS; RRMS—relapsing-remitting MS; SD—standard deviation; SPMS, secondary progressive MS. ^a^ mean value (standard deviation). ^ANOVA^ Analysis of variance. ^b^ median. ^c^ number of patients (%). ^Chi^ chi-squared test. ^H^ Kruskal-Wallis H test.

## Data Availability

The datasets generated and analyzed in the current study are available from the corresponding author upon reasonable request.

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
