# Peer review of "Vaccination Coverage against Tetanus, Diphtheria, Pertussis and Poliomyelitis and Validity of Self-Reported Vaccination Status in Patients with Multiple Sclerosis"

_jpm, 2022, doi:10.3390/jpm12050677_

Round 1

Reviewer 1 Report

The work presented by Langhorst et al. is an observational study on the vaccination status of patients with multiple sclerosis. The topic is interesting, considering the increasing prevalence of immunomodulatory therapies directed against humoral immunity for MS patients and the recent SARS-CoV-2 vaccination campaign.

Although it appears to be well conducted in a large population of patients, the study has an extremely limited impact on clinical practice for MS patients.

Indeed, the study focuses on vaccinations of lesser interest (since they are carried out in large prevalence given the strong recommendations of health authorities, mostly before MS diagnosis, and are mostly not considered by the data sheets of drugs licensed for MS) and does not consider the vaccinations that collect at the time the greatest interest in clinical practice, with a real impact in the management of people with MS (eg, vaccination against influenza virus, HBV, VZV, HPV, pneumococcus, meningococcus, SARS-CoV-2). Given this serious flaw in the study design, I recommend to not consider the manuscript suitable for publication in the Journal.

Author Response

Dear Reviewer,

we thank for the feedback on our manuscript. We agree with the reviewer that there are other vaccines that are of great interest in MS patients at this time. However, we chose these four vaccines because of the need for regular boosters and because they are well suited to investigate vaccination interest, the willingness of MS patients to get vaccinated and the current completeness of the vaccination status. We believe that research on other vaccinations can be informed by our study. We have added a paragraph starting at line 380 that refers to further vaccinations and necessary studies in the field.

To further emphasize the clinical relevance of our study, we have added a few sentences starting at lines 87, 92, 344 and 399. The special feature of our study is that, to the best of our knowledge, data on vaccination behavior, vaccination knowledge and vaccination status of MS patients were collected for the first time. Due to the increased risk of infections under DMD therapy, a complete vaccination status is of particular relevance for MS patients. The investigated vaccinations play an essential role in the clinical routine, since three of them (tetanus, diphtheria and pertussis) have to be refreshed every ten years and thus physicians have to keep an eye on the status of these vaccinations in their patients. Nevertheless, further research on other vaccinations is needed to get a more comprehensive view on the vaccination status in MS patients.

Please see the attachment with the revised version of the manuscript.

Reviewer 2 Report

Dear Authors,

First of all, I would like to tell you that I appreciate your work invested in this manuscript.
After careful reading I did not find sufficient scientific and clinical novelty in this manuscript to be published in this journal.
Submission of such paper would be more suitable to a journal with a local/regional character.

Author Response

Dear Reviewer,

we thank you for the time spent in evaluating our manuscript. To further emphasize the clinical relevance and novelty of our study, we have added a few sentences starting at lines 87, 92, 344 and 399. The special feature of our study is that, to the best of our knowledge, data on vaccination behavior, vaccination knowledge and vaccination status of MS patients were collected for the first time. Due to the increased risk of infections under DMD therapy, a complete vaccination status is of particular relevance for MS patients. The investigated vaccinations play an essential role in the clinical routine, since three of them (tetanus, diphtheria and pertussis) have to be refreshed every ten years and thus physicians have to keep an eye on the status of these vaccinations in their patients. Nevertheless, further research on other vaccinations is needed to get a more comprehensive view on the vaccination status in MS patients.

Please see the attachment with the revised version of the manuscript.

Reviewer 3 Report

The authors propose an interesting analysis of the documented and self-reported vaccination status of MS patients.

Associated demographic and clinical factors were also assessed. The study was conducted rigorously and is currently of great interest.

Author Response

Dear Reviewer,

we thank you for the positive comment and the interest in our study.

Please see the attachment with the revised version of the manuscript.

Reviewer 4 Report

The study aimed to examine the vaccination status (completeness), the validity of self-reported vaccination status, vaccination knowledge and vaccination behavior of MS patients. In particular the Authors investigated the vaccination status  for tetanus, diphtheria, pertussis and poliomyelitis. The topic is very interesting and actual. The finding that a large number of patients underwent a complete vaccination course for these pathologies (tetanus, diphtheria, pertussis and poliomyelitis) is comforting.

Nevertheless  the paper lacks information about the DMDs the PwMS were taking . This would have brought greater value. A comment should be added in discussion. 

Author Response

Dear Reviewer,

We thank you for the kind feedback on our manuscript. We added information about the DMDs for MS that were taken by the patients (results section, from line 184). In addition, we added a paragraph in the discussion from line 399 onwards, referring to the relevance of DMDs and possibly decreased immune response to vaccination under therapy.

Please see the attachment with the revised version of the manuscript.

Round 2

Reviewer 1 Report

The minor revisions made by the authors did not improve the overall interest and the potential impact of the study. 

Reviewer 2 Report

Dear Authors,

My previous decision was definite. Submission of such paper would be more suitable to a journal with a local/regional character.